# Groundwater Depth Overrides Tree-Species Effects on the Structure of Soil Microbial Communities Involved in Nitrogen Cycling in Plantation Forests

**Tiehang Wu [1],\*, Ashley Gray [1], Gan Liu [1], Hilary Kaminski [1], Bolanle Osi Efa [1] and Doug P. Aubrey [2,3]**

[1] Department of Biology, Georgia Southern University, Statesboro, GA 30460, USA; ag04507@georgiasouthern.edu (A.G.); gl01028@georgiasouthern.edu (G.L.); hk00894@georgiasouthern.edu (H.K.); bo00799@georgiasouthern.edu (B.O.E.)
[2] Savannah River Ecology Lab, University of Georgia, Aiken, SC 29802, USA; daubrey@srel.uga.edu
[3] Warnell School of Forestry and Natural Resources, University of Georgia, Athens, GA 30602, USA
\* Correspondence: twu@georgiasouthern.edu; Tel.: +1-912-478-5489

**Abstract:** Microbial communities found in soil ecosystems play important roles in decomposing organic materials and recycling nutrients. A clear understanding on how biotic and abiotic factors influence the microbial community and its functional role in ecosystems is fundamental to terrestrial biogeochemistry and plant production. The purpose of this study was to investigate microbial communities and functional genes involved in nitrogen cycling as a function of groundwater depth (deep and shallow), tree species (pine and eucalypt), and season (spring and fall). Soil fungal, bacterial, and archaeal communities were determined by length heterogeneity polymerase chain reaction (LH-PCR). Soil ammonia oxidation archaeal (AOA) *amo*A gene, ammonia oxidation bacterial (AOB) *amo*A gene, nitrite oxidoreductase *nrx*A gene, and denitrifying bacterial *nar*G, *nir*K, *nir*S, and *nos*Z genes were further studied using PCR and denaturing gradient gel electrophoresis (DGGE). Soil fungal and bacterial communities remained similar between tree species and groundwater depths, respectively, regardless of season. Soil archaeal communities remained similar between tree species but differed between groundwater depths in the spring only. Archaeal *amo*A for nitrification and bacterial *nir*K and *nos*Z genes for denitrification were detected in DGGE, whereas bacterial *amo*A and *nrx*A for nitrification and bacterial *nar*G and *nir*S genes for denitrification were undetectable. The detected nitrification and denitrification communities varied significantly with groundwater depth. There was no significant difference of nitrifying archaeal *amo*A or denitrifying *nir*K communities between different tree species regardless of season. The seasonal difference in microbial communities and functional genes involved in nitrogen cycling suggests microorganisms exhibit seasonal dynamics that likely impact relative rates of nitrification and denitrification.

**Keywords:** soil microbial community; nitrification; denitrification; nitrogen functional genes; biotic factor (pine and eucalypt tree species); abiotic factor (deep and shallow groundwater depth)

## 1. Introduction

Soil teems with diverse microorganisms, which regulate ecosystem functions such as decomposing organic materials and recycling nutrients [1]. Plant growth in terrestrial ecosystems is largely dependent on the activities of soil microorganisms [2–4]. Soil fungi have been recognized as contributing to degradation of recalcitrant lignocellulose complexes [5] and positively affecting soil aggregation [6,7]; therefore, they are important for maintaining high soil quality. Soil bacteria may regulate several key nutrient cycling processes including N, the essential nutrient for plant growth [3,8]. Archaea, once

thought to be restricted to extreme environments, have become a major contributor for N cycling processes in a wide variety of environments [9,10], including agricultural soils [11–14] and forest soils [15,16]. It is clear that soil microbes including fungi, bacteria, and previously undescribed archaea specialize in recycling nutrients and are the "backbone" to most biogeochemical cycles, especially nitrification and denitrification [14,17].

A variety of functional genes are involved in the nitrification and denitrification processes [8]. Both ammonia-oxidizing archaea (AOA) and ammonia-oxidizing bacteria (AOB) play roles in the process of nitrification [14,18]. The importance of either archaea or bacteria in the ammonia oxidation process under different environmental conditions has been reported [19,20]. Archaea are especially important for the nitrification processes under low-nutrient, low-pH, and sulfide-containing environments [20,21]. Two genes related to ammonia oxidation to nitrite, including *amo*A AOA and *amo*A AOB from archaea and bacteria, respectively, are the first and rate-limiting step in nitrification; thus, these genes are critical for the assessment of nitrification potential and communities [22]. Nitrite oxidoreductase genes (*nrx*A and *nrx*B) of nitrite oxidizers, such as nitrite-oxidizing bacteria (NOB), are responsible for completing the nitrification process by oxidizing nitrite to nitrate [23,24]. In contrast, denitrification is a chain of stepwise reduction processes of reducing nitrate ($NO_3^-$) to (1) nitrite ($NO_2^-$), (2) nitric oxide (NO), (3) nitrous oxide ($N_2O$), and (4) N gases ($N_2$). Each of the steps is regulated by the individual enzymes: nitrate reductase (*nar*), nitrite reductase (*nir*), nitric oxide reductase (nor), and nitrous oxide reductase (*nos*), respectively.

Plant species affect soil fungal community and diversity largely through root exudates [25,26]. Native plants have been reported to maintain unique soil fungal communities, whereas, non-native plants have less pronounced effects on soil fungal communities, as nonresident plant root exudates fail to support the native soil fungal community [25]. Soil water is essential for soil microorganisms and affects gas exchange and a variety of soil chemical reactions. Soil water contents affected by soil depth in semi-arid sites, precipitation, and drying–rewetting have been reported to influence soil microbial communities [27–31]. The reduced precipitation and consequently the decreased soil water content did not affect the bacterial community structure in general across the different forest management intensities in Germany [32]. Biotic and abiotic factors also affect the nitrification and denitrification communities. Soil water availability can influence nitrification, both in terms of process rates and community assemblage responsible for nitrification activity [33,34]. In two temperate forest soils, increased soil moisture rapidly and distinctly changed soil ammonia-oxidizing archaea (AOA) abundances [22]. Although exception may occur in some environments, plant species influences the fungal community structure more strongly, whereas soil moisture has larger impacts on bacterial communities in general [35].

Soil fungi, archaea, and bacteria, including functional groups of nitrification and denitrifying microbial communities, occupy different ecological niches [14,36]. It is critical to understand the effects of biotic and abiotic factors on microbial community and their underlying mechanisms in regulating nutrient cycling. Here, we studied the soil microbial communities and nitrogen functional genes regulating nitrification and denitrification processes within native loblolly pine (*Pinus taeda*) and non-native Eucalyptus tree plantations at the extreme ends of a groundwater depth gradient. Loblolly pine is the most commercially important tree species in the southeast of USA and is the primary candidate for bioenergy production. Eucalyptus is a genus containing over 700 species native to Australia that have been planted across the world for production forestry purposes. Compared with loblolly pine, Eucalyptus possesses deep root systems and is very well known for requesting a lot of water, with a huge capacity to get such water from groundwater in response to drought conditions [37,38]. The total soil fungal, bacterial, and archaeal communities together with the functional soil microbial genes, including nitrifying ammonia oxidation genes *amo*A AOA and *amo*A AOB, nitrite oxidation gene *nrx*A, and denitrifying bacterial genes *nar*G, *nir*K, *nir*S, and *nos*Z were used to understand how groundwater depth and tree species influenced soil microbial communities. We predicted that the bacterial community would be influenced more strongly by abiotic factors (i.e.,

groundwater depth and season), whereas the fungal community would be influenced more strongly by biotic factors (i.e., tree species).

## 2. Materials and Methods

### 2.1. Study Site Description and Sample Collection

The study site is located at the US Department of Energy's Savannah River Site in New Ellenton, SC, USA. The gradient of groundwater depth across the experimental watershed was planted with loblolly pine (*Pinus taeda*) in March 2013, an indigenous pine species to southeast USA, and Camden white gum (*Eucalyptus benthamii*) in October 2013, a non-native eucalypt. Each plot contains 168 individual trees with an area of 0.15 hectare. Three paired plots were located in deep groundwater (groundwater level > 10 m) and three were located in shallow groundwater (groundwater level < 2 m). Each of the six boxes (Figure 1 in red color) indicates the location of paired loblolly pine and eucalypt plots—three paired plots located in deep groundwater (U6, U7, and U10) and three paired plots located in shallow groundwater (U3, U5A, andU5B), which makes six repetitions for tree species and six repetitions of groundwater level. Soil type belongs to Fuquay loamy for plot U7 (https://soilseries.sc.egov.usda.gov/OSD_Docs/F/FUQUAY.html) and Dothan fine-loamy (https://soilseries.sc.egov.usda.gov/OSD_Docs/D/DOTHAN.html) for rest of the plots. Twenty soil cores around the two tree species from each plot were collected from top 20 cm of the soil layer and pooled together to obtain the soil samples in the spring (March) and the fall (August) of 2015 for this study.

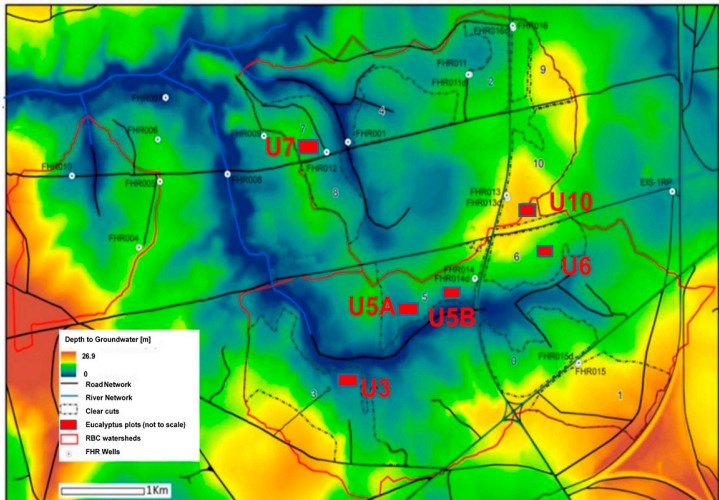

**Figure 1.** The varying water depths and the location of the plots where the samples were collected under loblolly pine (P), and eucalypt (E) fields located in deep groundwater (>10 m; U6, U7, and U10), and paired plots located in shallow groundwater (<2 m; U3, U5A, and U5B).

### 2.2. Soil Chemical Analysis

The chemical analysis was performed by A&L Plains Agricultural Laboratory, Inc. (Lubbock, TX, USA) and Environmental Microbiology Laboratory, Georgia Southern University (Statesboro, GA, USA). The components measured were organic matter (%), nitrate, phosphorus, potassium, magnesium, calcium, soil pH, and cation exchange capacity (A&L Plains Agricultural Laboratory), and soil water was analyzed according to the methods in Methods of Soil Analysis, Agronomy Book, No. 9 (Page et al. 1982, Madison, WI, USA). Soil ammonium was extracted with 2 M KCl and followed with the salicylate method [39].

### 2.3. DNA Extraction

DNA of soil samples was extracted using the PowerSoil DNA Isolation kit (Mobio Laboratories Inc., Carlsbad, CA, USA) following the manufacturer's instruction manual. All samples were rapidly

and thoroughly homogenized. The extracted DNA was stored at −20 °C to be used for PCR analysis and other downstream applications.

### 2.4. Length-heterogeneity PCR (LH-PCR)

Fungal internal transcribed spacer-1 (ITS-1) was amplified using forward primers NSI1 (5'-GATTGAATGGCTTAGTGAGG-3'), labeled with 6-FAM florescent dye, and reverse primer 58A2R (5'-CTGCGTTCTTCATCGAT-3'). Bacterial 16S rRNA gene was amplified from the extracted DNA using forward primers 27F (5'-AGAGTTTGATCMTGGCTCAG-3'), labeled with 6-FAM florescent dye, and reverse primer 355R (5'-GCTGCCTCCCGTAGGAGT-3'). Archaeal 16S rRNA gene was amplified with primer pair of Ar3f (5'-TTCCGGTTGATCCTGCCGGA-3') and PARCH519R_mod (5'-TTACCGCGGCGGCTG-3'), labeled with HEX fluorescent dye [40]. Length heterogeneity of amplified PCR products was assessed on Applied Biosystems 3500 genetic analyzer (Applied Biosystems, Foster City, CA, USA). The size of each fragment was assigned to operational taxonomic units (OTUs) using a ±0.5 base pair (bp) criterion [41,42].

### 2.5. Denaturing Gradient Gel Electrophoresis (DGGE) Analysis

PCR amplification of different nitrification and denitrification genes (*amo*A AOA, *amo*A AOB, *nrx*A, *nar*G, *nir*K, *nir*S, *nos*Z genes) was performed using GoTaq® Green Master Mix 2× (Promega, Madison, WI, USA) by corresponding primers (Table 1). A 25 μL volume of each reaction for each soil DNA extraction was amplified using a Master Cycler (Eppendorf, Hamburg, Germany). The 25 μL PCR reaction included 13.5 μL of GoTaq® Green Master Mix 2×, 0.81 μL of forward and reverse primers (10 μM), 2.7 μL of BSA (10 mg mL$^{-1}$), 1.5 μL of DNA template and 6.13 μL of nuclease-free water. The GC clamp was added to the corresponding primer position as indicated by star symbol (*) in Table 1. The PCR conditions for each gene were according to the procedures as described in the original references as listed in Table 1 [18,24,43–50].

**Table 1.** Primer sequence pairs used to amplify *amo*A ammonia oxidation archaeal (AOA), *amo*A ammonia oxidation bacterial (AOB), *nrx*A, *nar*G, *nir*K, *nir*S, and *nos*Z genes for DGGE (with GC clamp) and Q-PCR (without GC clamp).

| Gene | Primer | Primer Sequences | References |
|---|---|---|---|
| *amo*A AOA | Arch-*amo*AF * <br> Arch-*amo*AR | 5'-(C/G)TAATGGTCTGGCTTAGACG-3' <br> 5'-GCGGCCATCCATCTGTATGT-3' | [18] |
| *amo*A AOB | *amo*A-1F * <br> *amo*A-2R | 5'-GGGGTTTCTACTGGTGGT-3' <br> 5'-CCCCTC(G/T)G(C/G)AAAGCCTTCTTC-3' | [45] |
| *nrx*A | F1370 F1 *nrx*A <br> F2843 R2 *nrx*A * | 5'-CAGACCGACGTGTGCGAAAG-3' <br> 5'-TCCACAAGGAACGGAAGGTC-3' | [24] |
| *nar*G | *nar*G2179F <br> *nar*G2488R * | 5'-TAC(A/T)T(C/G)CT(C/G)AAGTACCT(C/G)CT-3' <br> 5'-C(C/G)TTGTAGATCTCCCA(A/G)TC-3' | [48] <br> [49] |
| *nir*K | F1aCu <br> R3Cu * | 5'-ATCATGGT(C/G)CTGCCGCG-3' <br> 5'-GCCTCGATCAG(A/G)TTGTGGTT-3' | [50] |
| *nir*S | cd3aF <br> R3cd * | 5'-GT(C/G) AAC GT(C/G)AAG GA(A/G)AC(C/G)GG-3' <br> 5'-GA(C/G)TTCGG(A/G)TG(C/G)GTCTTGA-3' | [47] |
| *nos*Z | *nos*Z-F <br> *nos*Z1622R * | 5'-CG(C/T)TGT TC(A/C)TCGACAGCCAG-3' <br> 5'-CGC (G/A)A(C/G)GGCAA(C/G)AAG GT(C/G)CG-3' | [46] |

* Indicates that a GC-clamp was added to the primer for PCR reaction of DGGE.

Amplified gene products (*amo*A AOA, *amo*A AOB, *nrx*A, *nar*G, *nir*K, *nos*Z, and *nir*S) were run on 8% (w/v) acrylamide with a linear chemical gradient ranging from 4%–70% using a DGGEK-1001 Cipher DGGE Kit (C.B.S. Scientific, Del Mar, CA, USA). Gels were run in 1× Tris-acetate-EDTA (TAE) buffer at 120 V for 9 h, with temperature of 60 °C for all gene products. After electrophoresis, gels

were stained with 1× SYBR Gold Nucleic Acid Gel Stain (Molecular Probes, Inc., Eugene, OR, USA) by carefully oscillating the gel in the stain for 20 min. Using UVP Benchtop UV Transilluminators, gels were illuminated and captured. Obtained DGGE band patterns were normalized following the function of the GelComparII software (Applied Maths, Austin, TX, USA) and band matching were conducted. The quantitative data of DGGE bands were exported for further nonparametric multivariate analyses of soil microbial communities (see statistical analysis of fragments of microbial DNA in Section 2.7).

## 2.6. Quantitative PCR (Q-PCR)

Quantitative PCR was performed using the QuantStudio™ 6 Flex Real-Time PCR System (Life Technologies, Carlsbad, CA, USA). Quantification utilized the fluorescent dye SYBR-Green I which binds to double-stranded DNA during amplification. Primer details and sequences for the target genes are listed in Table 1. All PCR mixtures contained the recommended 25 μL protocol consisting of 12.5 μL of 2× GoTaq® Colorless Master Mix (Promega, Madison, WI, USA), 0.5 μL of 10 μM forward and reverse primer, 1 μL BSA, 2 μL SYBR® of 1×, 6 μL of nuclease-free water, and 2 μL of DNA template. The Q-PCR of each sample was run in triplicate to control for mechanical and technical errors. The mean of all three DNA quantities was used for statistical analysis.

## 2.7. Statistical Analysis

Univariate statistics: Treatment effects (groundwater depth, tree species, and season) on soil chemical characteristics were analyzed using analysis of variance (ANOVA). The homogeneity of variances and normality of distribution were tested with the Levene and Kolmogorov–Smirnov tests, respectively. In addition, the non-normally distributed dependent variables were log10-transformed before analysis. All data were analyzed by three-way ANOVA on groundwater depth, tree species, and season with means and a post-hoc method (Duncan's method) for multiple comparisons at a 5% significance level. All the statistical analyses were performed with JMP® Pro 12.1.0 (SAS Institute Inc., Cary, NC, USA).

Statistical analysis of fragments of microbial DNA: Nonparametric multivariate analyses procedures including pairwise Bray–Curtis similarity of LH-PCR fragment profiles and band matrices of DGGE obtained above were applied for microbial community analysis. A square-root transformation was applied to the data before construction of the similarity matrices. Cluster analysis was performed to compare the similarity microbial and functional gene communities; and analysis of similarity (ANOSIM) and permutational multivariate analysis of variance (PERMANOVA) including principal coordinates analysis (PCO) with Spearman's correlations of variables with the PCO axes were applied to test the significance of microbial community difference and its correlation with environmental factors. A significance level of 5% for ANOSIM (ANOSIM $p < 0.05$) was applied to detect a difference of microbial community between groups. The correlations between nitrogen functional genes and environmental factors were determined by ENVIRO-BEST function. All nonparametric multivariate analyses procedures, including calculation of Bray–Curtis similarity matrices, cluster analysis ANOSIM, PERMANOVA, and PCO and ENVIRO-BEST were conducted using the PRIMER-E & PERMANOVA plus statistical (PRIMER-E Ltd., Plymouth Marine Laboratory, Plymouth, UK) software.

## 3. Results

### 3.1. Soil Chemical Characteristics

Groundwater depth, tree species, and season differently affected the soil chemical characteristics of soil organic matter (%), pH, phosphorus, potassium, magnesium, and nitrate content, soil CEC, as well as water content (%) (Table 2). Soil at the site of deep groundwater had significantly higher soil pH and lower water content (%) than at shallow groundwater depth ($p < 0.05$). Eucalypt trees on the plots significantly increased soil organic matter (%), as well as soil water content (%), CEC, phosphorus and magnesium contents, compared with pine field plots ($p < 0.01$). The spring samples had significantly higher nitrate, potassium, magnesium, and CEC contents than in the fall samples ($p <$

0.05), and the nitrate concentration was almost four times higher in the spring than in the fall (Table 2). However, significantly lower soil phosphorus contents were observed in the fall than in spring. No interactions ($p > 0.05$) among groundwater depth, tree species, and season were observed for all soil organic matter, nitrate, potassium, and soil CEC, but interaction between tree species and season was observed for soil phosphorus and magnesium contents (Table 2).

## 3.2. Soil fungal Communities

Soil fungal communities determined by LH-PCR in the spring significantly differed from soil fungal communities in the fall (ANOSIM $p < 0.01$, Figure 2A). For the spring samples, tree species significantly affected the soil fungal community structure (ANOSIM $p = 0.01$, Figure 2A,B), whereas no significant effect (ANOSIM $p = 0.13$) of groundwater depth on soil fungal community structure was detected. For the fall samples, the loblolly pine and eucalypt also significantly affected the soil fungal community structure (ANOSIM $p < 0.01$, Figure 2A,C), but no significant effect of groundwater depth on soil fungal community structure was detected (ANOSIM $p = 0.69$). The effect of environmental factors on soil fungal community structure can be seen in the PCO ordination with the length and direction of each vector through the relationship between the variable and the PCO axes, where soil organic matter (%) had a strong negative relationship with PCO1 to the separation of soil fungal communities in both the spring (Figure 2B) and the fall (Figure 2C), whereas soil ammonium (Figure 2B) and soil pH (Figure 2C) positively contributed to the different soil fungal communities in the spring and fall with PCO1, respectively.

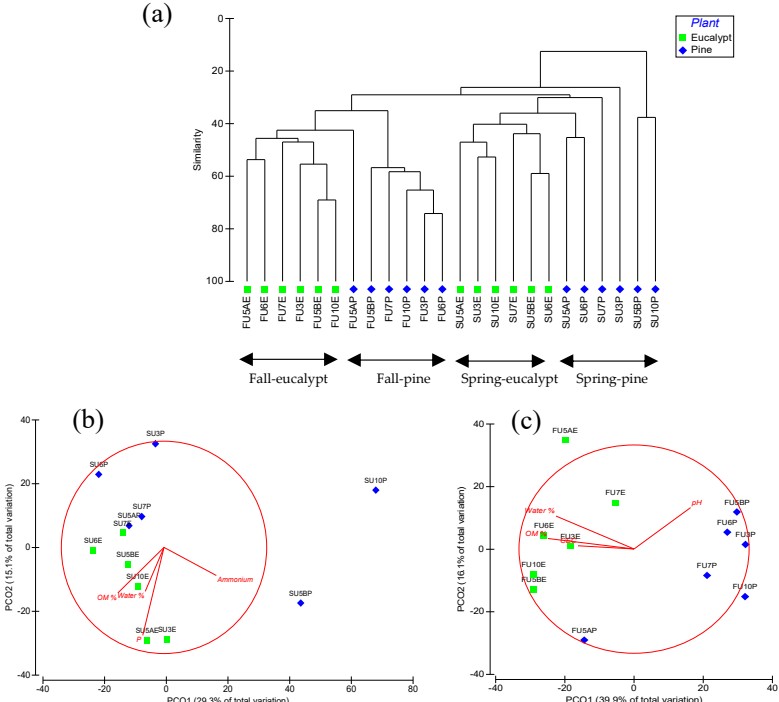

**Figure 2.** (**a**) Cluster analysis of the spring (S) and fall (F) samples. (**b**,**c**) Ordination of environmental variables using PCO on the (b) spring and (c) fall samples, with a vector overlay showing environmental factor with vectors (Spearman's correlations of variables with the PCO axes) using LH-PCR fragments amplified with NSI1/58A2R primer pair for fungal ITS1 under the loblolly pine (P) and eucalypt (E) plots at the sites with deep groundwater (>10m; U6, U7, and U10) and shallow groundwater (<2 m; U3, U5A, and U5B). Only the environmental factors that have an absolute correlation $\rho$ value among the top four rankings (the circle represents a correlation value of $\rho = 1.0$) are shown. The sample names beginning with "S" and "F" are the samples collected from the spring and fall, respectively.

**Table 2.** Soil chemical characteristics and the significance (*p*-value) of statistical analysis under different groundwater depth, tree species and season. Means ± SE (*n* = 3) are presented. Soils were collected from top 20 cm of soil layer. Means for main effects followed by different letters are significantly different at *p* < 0.05.

| Effects and Interactions | | Soil Organic Matter (%) * | Soil pH ** | Soil Water (%) | Ammonium (ppm) | Nitrate (ppm) | Phosphorus (ppm) | Potassium (ppm) | Magnesium (ppm) | Calcium (ppm) | CEC (meq/100 g) |
|---|---|---|---|---|---|---|---|---|---|---|---|
| Groundwater | Deep | 2.0 ± 0.3 a | **4.2 ± 0.2 a** | **11.6 ± 4.3 b** | 0.6 ± 0.2 a | 26.9 ± 7.8 a | 80.5 ± 11.2 a | 34.4 ± 4.9 a | 34.2 ± 9.1 a | 116.6 ± 94.2 a | 4.0 ± 0.8 a |
| | Shallow | 2.4 ± 0.3 a | **3.8 ± 0.1 b** | **17.9 ± 6.5 a** | 1.0 ± 0.5 a | 33.0 ± 9.5 a | 96.0 ± 18.2a | 37.8 ± 8.5a | 42.4 ± 11.9 a | 300.0 ± 134.7 a | 5.1 ± 1.2 a |
| Tree | Eucalypt | **2.9 ± 0.3 a** | 4.1 ± 0.2 a | **18.3 ±5.9 a** | 0.6 ± 0.1 a | 29.3 ± 7.7 a | **111.1 ±18.7 a** | 42.8 ± 8.9 a | **54.5 ± 13.0 a** | 319.9 ± 133.9 a | **6.3 ± 1.3 a** |
| | Pine | **1.5 ± 0.1 b** | 3.9 ± 0.1 a | **11.3 ± 4.4 b** | 1.0 ± 0.5 a | 30.7 ± 9.7 a | **65.4 ± 4.9 b** | 29.3 ± 3.0 a | **22.1 ± 3.4 b** | 96.7 ± 14.9 a | **2.9 ± 0.4 b** |
| Season | Spring | 2.4 ± 0.2 a | 4.1 ± 0.2 a | 14.3 ± 4.9 a | 0.7 ± 0.2 a | **47.6 ± 9.7 a** | 65.0 ± 6.1 b | **47.2 ± 8.3 a** | **57.2 ± 12.5 a** | 327.8 ± 132.9 a | **6.3 ± 1.3 a** |
| | Fall | 2.1 ± 0.3 a | 3.9 ± 0.1 a | 15.7 ± 7.6 a | 0.9 ± 0.5 a | **12.3 ± 1.6 b** | 111.5 ± 18.3 a | **25.0 ± 8.2 b** | **19.4 ± 2.9 b** | 88.8 ±12.4 a | **2.9 ± 0.4 b** |
| **Significance level** | | | | | | | | | | | |
| Groundwater | | 0.2736 | **0.0443** | **0.0012** | 0.4336 | 0.5830 | 0.2249 | 0.6894 | 0.4339 | 0.1300 | 0.2929 |
| Tree | | **0.0006** | 0.6223 | **0.0004** | 0.4495 | 0.8978 | **0.0078** | 0.1180 | **0.0058** | 0.0696 | **0.0045** |
| Season | | 0.3406 | 0.5369 | 0.5731 | 0.7148 | **0.0048** | **0.0089** | **0.0150** | **0.0019** | 0.0537 | **0.0043** |
| Groundwater × Tree | | 0.5380 | 0.9121 | 0.5244 | 0.7722 | 0.8150 | 0.1814 | 0.6039 | 0.3410 | 0.2485 | 0.3923 |
| Groundwater × Season | | 0.8624 | 0.0976 | 0.7863 | 0.3169 | 0.4149 | 0.0704 | 0.5500 | 0.4810 | 0.1479 | 0.3363 |
| Tree × Season | | 0.4465 | 0.3769 | 0.0555 | 0.3183 | 0.8267 | **0.0083** | 0.0651 | **0.0398** | 0.1323 | 0.0519 |

* Soil organic matter (%) was determined as soil organic carbon % using Walkley–Black Method. ** The significance level (*p*-value) of soil pH is log10 transformed. The bold fonts indicate significance at *p* < 0.05.

### 3.3. Soil Bacterial and Archaeal Communities

Soil bacterial communities determined by LH-PCR in the spring were significantly different from soil bacterial communities in fall (ANOSIM $p < 0.01$, Figure 3A). Soil bacterial communities in the spring samples (Figure 3A,B) differed between groundwater depths (ANOSIM $p = 0.06$) but remained similar between plant species (ANOSIM $p = 0.89$). Soil bacterial communities in the fall samples (Figure 3A,C) also differed between groundwater depths (ANOSIM $p = 0.02$) but remained similar between plant species (ANOSIM $p = 0.85$). The PCO ordination indicated that water content (%) had a positive relationship with PCO1, whereas soil pH negatively contributed to the separation of soil bacterial communities in the spring but contributed positively in the fall with PCO2 (Figure 3B,C).

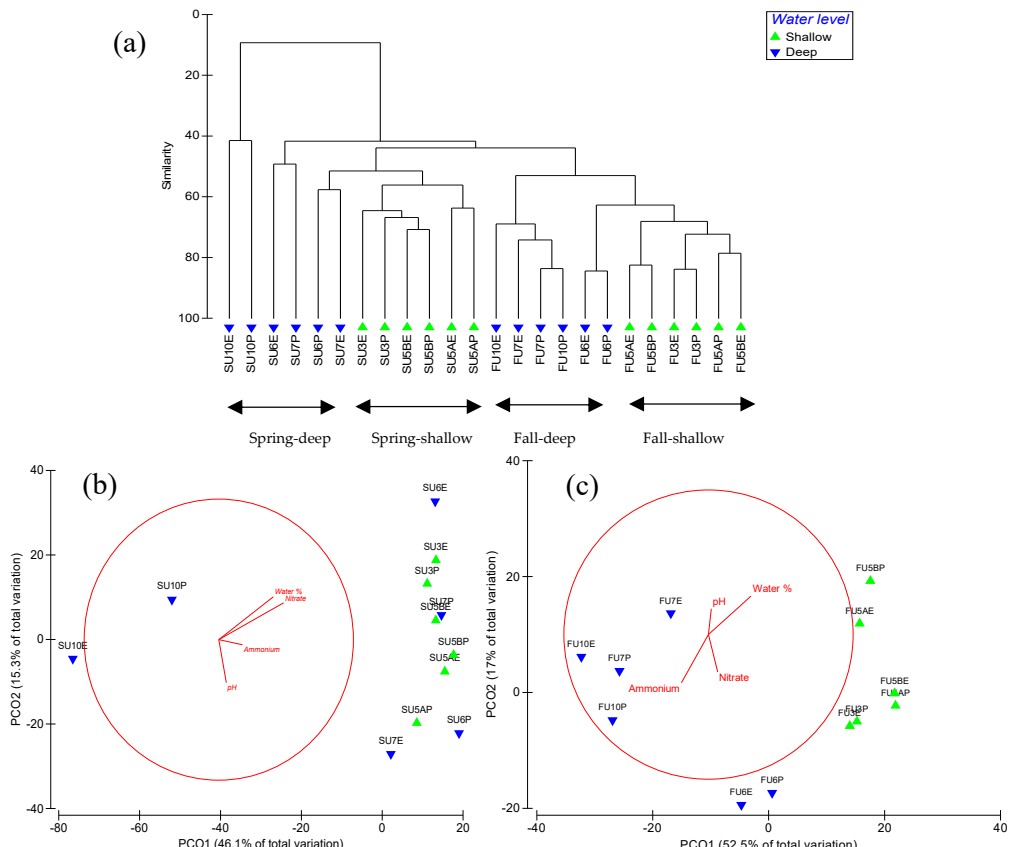

**Figure 3.** (**a**) Cluster analysis of the spring (S) samples and fall (F) samples of 2015. (**b,c**) Ordination of environmental variables using PCO on the (**b**) spring and (**c**) fall samples, with a vector overlay showing environmental factor with vectors (Spearman's correlations of variables with the PCO axes) using LH-PCR fragments amplified with 27F/355R primer pair for bacterial 16S rRNA under the loblolly pine (P) and eucalypt (E) plots at the sites with deep groundwater (>10 m; U6, U7, and U10) and shallow groundwater (<2 m; U3, U5A, and U5B). Only the environmental factors that have an absolute correlation $\rho$ value among the top four rankings (the circle represents a correlation value of $\rho = 1.0$) are shown. The sample names beginning with "S" and "F" are the samples collected from the spring and fall, respectively.

Soil archaeal communities determined by LH-PCR in the spring were significantly different from soil archaeal communities in fall (ANOSIM $p < 0.01$, Figure 4A). Soil archaeal communities in the spring samples (Figure 4A,B) differed between groundwater depths (ANOSIM $p = 0.02$) but remained similar between plant species (*ANOSI*M $p = 0.85$). Soil archaeal communities in the fall samples (Figure 4A,C) were not significantly affected by neither groundwater depth (ANOSIM $p = 0.20$) nor plant species (ANOSIM $p = 0.42$). Soil pH, water content (%), and ammonium contents all positively contributed to

the different soil archaeal communities with PCO1 in both the spring and fall samples (Figure 4B,C). Soil nitrate contents positively and negatively contributed to the different soil archaeal communities with PCO1 in the spring and fall samples, respectively (Figure 4B,C).

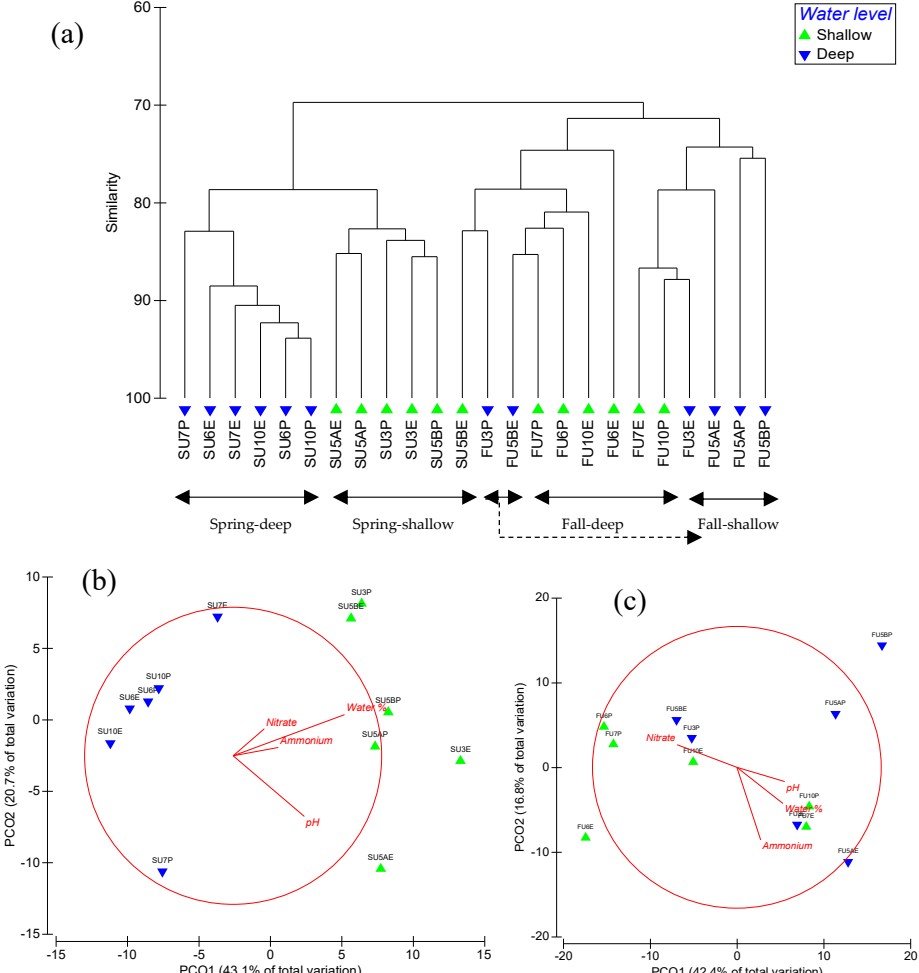

**Figure 4.** (**a**) Cluster analysis of the spring (S) samples and fall (F) samples of 2015. (**b**,**c**) Ordination of environmental variables using PCO on the (b) spring, and (c) fall samples, with a vector overlay showing environmental factor with vectors (Spearman's correlations of variables with the PCO axes) using LH-PCR fragments amplified with Ar3f/PARCH519R_mod primer pair for archaeal 16S rRNA under the loblolly pine (P) and eucalypt (E) plots at the sites with deep groundwater (>10 m; U6, U7, and U10) and in shallow groundwater (<2 m; U3, U5A, and U5B). Only the environmental factors that have an absolute correlation $\rho$ value among the top four rankings (the circle represents a correlation value of $\rho$ = 1.0) are shown. The sample names beginning with "S" and "F" are the samples collected from the spring and fall, respectively.

### 3.4. Soil Nitrification Archaea and Bacteria

There was a significantly different soil nitrification community determined by DGGE with ammonia-oxidizing archaea *amo*A AOA gene (Table 3, Figure 5A,B) at shallow groundwater compared with deep groundwater in the spring (ANOSIM $p = 0.02$), but not in the fall (ANOSIM $p = 0.58$). The communities of *amo*A AOA gene were significantly different between the spring and the fall (ANOSIM $p = 0.01$). Tree species did not influence the communities of ammonia-oxidizing archaea *amo*A AOA gene in either the spring (ANOSIM $p = 0.68$) or the fall (ANOSIM $p = 0.86$). ENVIRO-BEST analysis indicated that nitrate, Mg, and Ca contributed to the observed communities of ammonia-oxidizing

archaea *amo*A AOA gene with correlation coefficient of >10% (Table 4). Soil nitrification communities of ammonia-oxidizing bacteria *amo*A AOB were not detected by DGGE in the soil samples (Table 3).

**Table 3.** The significance (*p*-values) of ANOSIM analyses on DGGE for *amo*A AOA, *amo*A AOB, *nrx*A, *nar*G, *nir*K, *nir*S, and *nos*Z genes under deep and shallow groundwater depths of pine and eucalypt plantation in the spring and fall.

| Gene | | Season | Water Level | | Plant Species | |
|---|---|---|---|---|---|---|
| | | | Spring | Fall | Spring | Fall |
| Nitrification | *amo*A AOA | 0.01 * | 0.02 * | 0.58 | 0.68 | 0.86 |
| | *amo*A AOB | nd | nd | nd | nd | nd |
| | *nrx*A | nd | nd | nd | nd | nd |
| Denitrification | *nar*G | nd | nd | nd | nd | nd |
| | *nir*K | 0.01 * | 0.04 * | 0.22 | 0.46 | 0.63 |
| | *nir*S | nd | nd | nd | nd | nd |
| | *nos*Z | 0.01 * | 0.83 | 0.32 | 0.61 | 0.88 |

The numbers with * indicate significantly different communities ($p < 0.05$) by ANOSIM analysis. The nd indicates that genes were not detected by regular PCR amplifications.

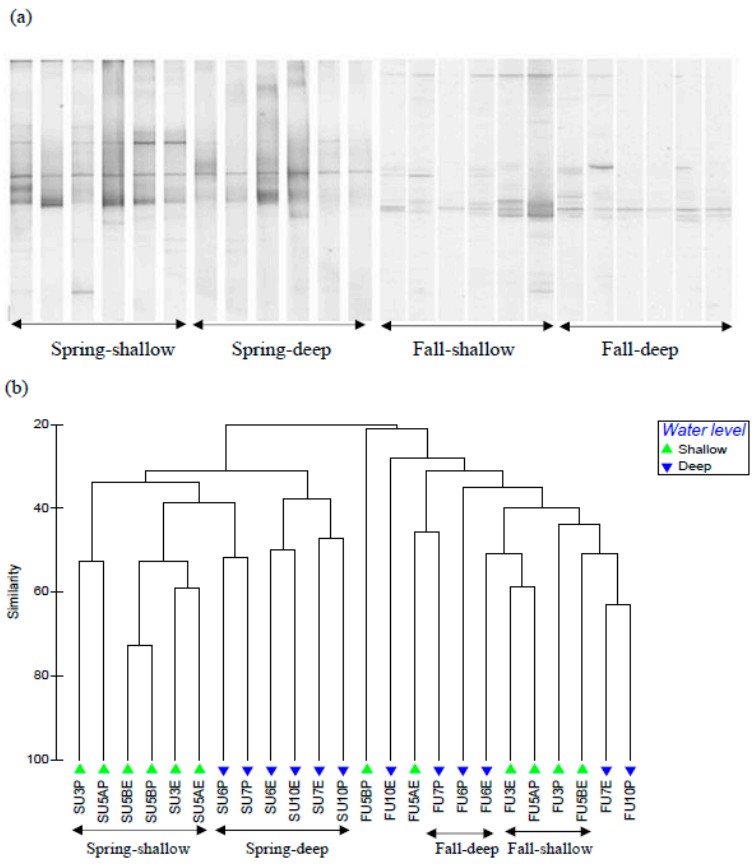

**Figure 5.** (**a**) DGGE profiles and (**b**) cluster analysis of nitrification archaeal *amo*A AOA gene using Arch-*amo*AFa/Arch-*amo*AR primer pair from the spring and fall samples of eucalypt and pine under deep (spring–deep, fall–deep) and shallow (spring–shallow, fall–shallow) groundwater depths.

**Table 4.** The ENVIRO-BEST data for the correlation between *amo*A AOA, *nir*K, and *nos*Z genes and environmental factors under deep and shallow groundwater depths of pine and eucalypt plantation in the spring and fall.

| Genes | Variables | Correlation * |
|-------|-----------|---------------|
| *amo*A AOA | Nitrate (ppm) | 17.4% |
| | Mg (ppm) | 11.5% |
| | Ca (ppm) | 10.4% |
| *nos*Z | All less than 10.0% | |
| *nir*K | Nitrate (ppm) | 16.6% |
| | Mg (ppm) | 14.0% |
| | K (ppm) | 13.1% |
| | Water (%) | 10.7% |

* Only environmental variables with a correlation value >10%, determined by ENVIRO-BEST function, are included in the table.

The nitrite oxidoreductase gene *nrx*A of nitrite oxidizer was not detected by regular DGGE procedures using extracted soil DNA, but it was detected in the samples when a two-step nested PCR reaction for DGGE was applied. Q-PCR indicated that threshold cycles (CT) of *nrx*A gene were significantly shortened from 26 to 28 cycles (with low DNA amounts) of regular PCR to only around 10 cycles (with high DNA amounts) of two-step nested PCR ($p < 0.05$).

*3.5. Soil Denitrification Bacteria*

In the spring samples, obvious bands of *nos*Z gene products were detected by DGGE in ten of twelve samples. We did not detect bands in a sample from pine grown in shallow groundwater (U3P) and another in deep groundwater (U6P). Based on ANOSIM analyses of the *nos*Z gene obtained from the 10 samples (Figure 6A,B), the communities of *nos*Z gene were significantly different between the spring and the fall (ANOSIM $p < 0.05$). There was no significantly different soil denitrification community under the shallow groundwater depth sites compared with deep groundwater depth sites (ANOSIM $p = 0.83$, Table 3); no difference between tree species (ANOSIM $p = 0.61$, Table 3) was observed either in the spring samples. In the fall samples, there were no significantly different soil denitrification communities of *nos*Z gene under the shallow groundwater sites compared with the deep groundwater sites (ANOSIM $p = 0.32$, Table 3). Likewise, no difference was observed between tree species (ANOSIM $p = 0.88$, Table 3) either. ENVIRO-BEST analysis indicated that correlation coefficients of *nos*Z gene with all of the environmental factors were <10% (Table 4).

There was a significantly different soil denitrification community of *nir*K gene detected by DGGE under the shallow groundwater depth sites compared with deep groundwater depth sites in both the spring (ANOSIM $p = 0.04$) and the fall (ANOSIM $p = 0.02$) samples (Figure 7A,B, Table 3). No difference of denitrifying bacterial community of *nir*K gene was observed between different plant species in either the spring (ANOSIM $p = 0.63$) or the fall (ANOSIM $p = 0.46$) samples (Figure 7A,B, Table 3). Likewise, the communities of *nir*K gene were significantly different between the spring and the fall (ANOSIM $p = 0.01$). ENVIRO-BEST analysis indicated that nitrate, Mg, K, and water content (%) contributed to the observed communities of *nir*K gene with correlation coefficients of >10% (Table 4).

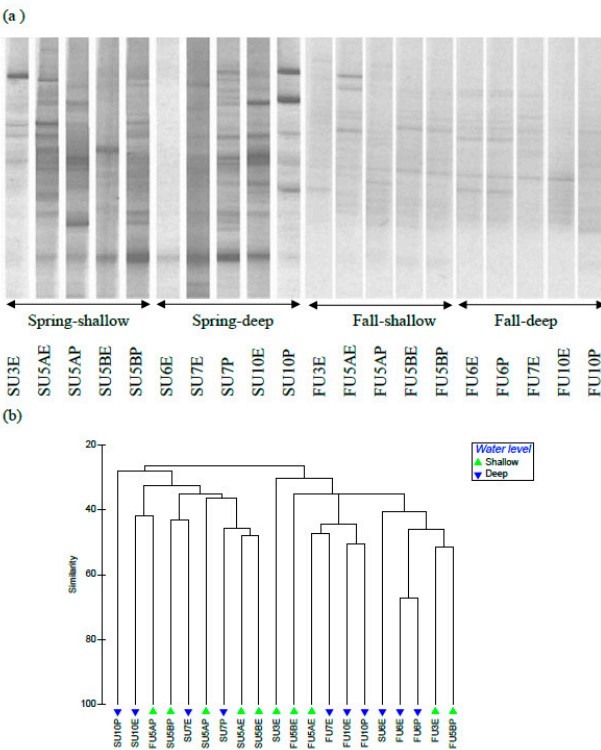

**Figure 6.** (**a**) DGGE profiles and (**b**) cluster analysis of bacterial denitrification *nos*Z gene using *Nos*ZF/*Nos*Z1622R primer pair from the spring and fall samples of eucalypt and pine under deep (spring–deep, fall–deep) and shallow (spring–shallow, fall–shallow) groundwater depths.

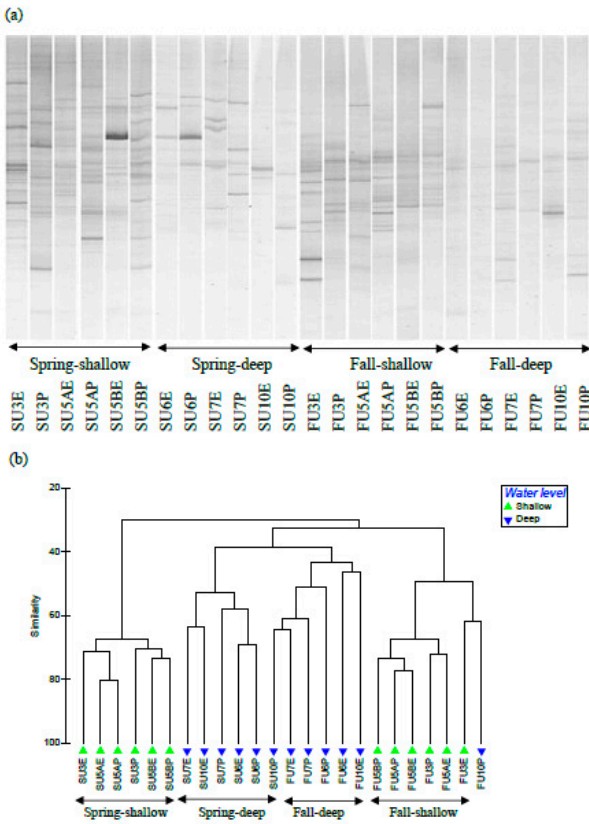

**Figure 7.** (**a**) DGGE profiles and (**b**) cluster analysis of denitrification *nir*K gene using F1aCu/R3Cu-GC primer pair for denitrifying bacteria from the spring and fall samples of eucalypt and loblolly pine under deep (spring–deep, fall–deep) and shallow (spring–shallow, fall–shallow) groundwater depths.

## 4. Discussion

Both biotic and abiotic factors influenced the diversity and functional genes of soil microbial communities. As we predicted, the soil fungal communities were affected more strongly by tree species, whereas soil bacterial and archaeal communities were influenced more strongly by groundwater depth. Different plant species and their root exudates have been reported to affect soil microbial community and diversity [25,51,52]. Plant root exudates and soil organic matter content play an underappreciated role in shaping the soil fungal communities [25,26]. It is demonstrated that beech and spruce trees species, with different litter quality, selected different soil fungal communities expressing different set of genes involved in organic matter degradation [52]. Another study indicated that 35%–37% of the dominant fungal "species" (operational taxonomic units, OTUs) were restricted to one or two tree species, whereas only about 15%–45% of fungal OTUs were common under six or seven tree species in the study [53], suggesting the significant effects of plant species on fungal community compositions. The research of our study indicated that tree species of eucalypt significantly increased soil organic matter (%), soil phosphorus, magnesium, soil water content (%) and CEC over tree species of pine, and the interaction between tree and season for phosphorus and magnesium was also detected (Table 2), suggesting that the different root growth and activities with season as well as different characteristics of pine and eucalyptus roots could affect the nutrient uptake differently, therefore leading to the observed interaction on soil phosphorus and magnesium concentration. Furthermore, the difference of the root growth and activities with seasonal change could affect the nutrient uptakes differently and lead to different nutrient concentration and CEC with season, thus perhaps contributing to significant differences in soil fungal community.

Soil water content and availability were related to decomposition and microbial growth as well as to changes in the microbial community structure [54]. Our study suggested that groundwater depths affected soil bacterial communities in both spring and fall and affected soil archaeal communities and functional genes of N cycling in the spring. Significantly higher soil pH but lower soil water content (%) were observed under deep than shallow water depths (Table 2). The deep root systems of a Eucalyptus tree requesting a lot of water, with a huge capacity to get such water from groundwater, may greatly affect soil water content (%), total soil bacteria, and nitrifying/denitrifying bacterial communities observed in this study. The spring and fall water contents were not significantly different in the collected soil samples of two sampling times in the present study, however, the seasonal fluctuation of the water content and the effects of anoxic conditions of the soils for the effect of anaerobic soils may cause the seasonal difference between the functional genes. Study on the relationship of soil water contents and microbial functional genes indicated that increased soil moisture rapidly and distinctly changed soil AOA abundances in two temperate forest soils [22]. However, contrary to our present study, it was also observed that fungal and actinobacterial community changes due to short- and long-term water-level changes at three different sites in a boreal peatland complex, indicating that fungal community responds to persistent water-level drawdown, whereas actinobacterial community was less sensitive to hydrological change of short-term groundwater depth drawdown [55]. The interaction between abiotic and biotic factors affects microbial community structure in the natural environment, with effects of plant species on fungal community structure being statistically significant; effects of moisture on bacterial community structure were also significant [35,52].

Functional genes regulating the N cycle, including nitrification and denitrification, have important implications for N-use efficiency in agricultural and forest ecosystems as well as for environmental quality. Nitrification in soil converts relatively immobile ammonium-N to nitrite first and then to highly mobile nitrate-N. Soil matrix, water status, aeration, temperature, and pH have strong influence on nitrification rates [56]. Genes involved in the nitrification include *amo*A AOA and *amo*A AOB from archaea and bacteria, respectively. Groundwater depth significantly changed the *amo*A AOA community in this study, indicating that the groundwater depths and consequent changes of soil pH are the major factors influencing nitrification. The *amo*A AOB was not detected in this study. The importance of archaea or bacteria in the process of nitrification has been observed in agricultural

soils [19,20]. The contribution of bacteria and archaea to nitrification processes in soil are affected by soil pH. In acidic soil conditions, archaea may be more important than bacteria for the ammonium oxidation. Another study indicated that AOA might influence the nitrogen cycle in low-nutrient, low-pH, and sulfide containing environments [21]. The soil pH in our study site was acidic, ranging from 3.4 to 6.0 in raw data of pH value, confirming that the archaeal *amo*A AOA, but not bacterial *amo*A AOB, dominated in the nitrification process in low-pH soil. The soil pH at the site with shallow groundwater depth was significantly lower than that at the site with deep groundwater depth in this study (Table 2), which may explain the observed dominance of *amo*A AOA in the study sites as well as different communities of nitrification functional genes under deep and shallow groundwater depths.

Denitrification is closely related to loss of nitrogen, when dinitrogen is the dominant product, or to an environmental concern as the intermediate product $N_2O$ of denitrification is not only harming the ozone layer but also a potent greenhouse gas [8,50]. Denitrification is known to be influenced by temperature, soil carbon, and oxygen conditions of soil controlling aerobic and/or anaerobic microbial processes [57,58]. Because many factors are involved in the processes, denitrification activities may not always correlate with the denitrifying microbial biomass and community structure found in each environment. Using the DGGE for studying the denitrification functional genes, the *nir*K and *nos*Z were determined as the most abundant denitrifying functional genes, and the *nir*K gene performed unique community structure under different groundwater depths. Similar to total bacterial 16S rRNA gene and nitrification *amo*A AOA gene, the *nir*K denitrifying gene was also affected by groundwater depth, but not tree species. This is consistent with a recent study that water content differences have stronger effects than plant functional groups on soil bacteria in a steppe ecosystem [59], and *nir*K gene abundance rapidly increased in response to wet conditions until substrate ($NO_3^-$) became a limiting factor [60]. In contrast, abundant denitrifying *nos*Z gene was detected, but it was not affected by either groundwater depth or plant species. Previous studies indicated that denitrifying functional genes interacted with complex soil environment, including soil C, moisture, N contents, etc., and less is known about how the genes are influenced by soil environment or affect the denitrification rate [8]. The abundance of denitrification genes had been reported as significantly different in forest and agricultural sites. A study indicated that the *nos*Z gene was significantly dominant in forest sites, while *nir*S gene was more abundant than *nos*Z gene in agricultural sites [61]. Our study also confirmed the dominant *nos*Z gene but undetected *nir*S gene in the forest site of present study. The intense bands of *nos*Z gene were detected across soil samples in this study but did not vary with respect to fluctuating groundwater depths or tree species. Differences in microbial communities with respect to groundwater depths on *nos*Z gene may have been affected by other factors, such as temperature, redox potential, and soil pH. Some studies have shown a positive correlation between denitrifying genes abundance and denitrification activity, while others have shown quite the opposite [62]. No correlation between groundwater depths and soil microbial community of *nos*Z gene suggested that the abundance of denitrifying *nos*Z gene was independent of denitrifying activity across groundwater depth. There have been several past studies supporting this claim. For example, *nir*K and *nir*S positively correlate in abundance with oxygenation, and *nos*Z positively correlates with soil pH [63]. However, significant difference in soil microbial community related to *nir*K genes was observed across the groundwater depth gradient in samples of both spring and fall, and ENVIRO-BEST analysis suggested that *nir*K genes were related to soil nitrate, Mg, K, and water content (%). Soil microbial communities with various functional genes related to denitrifying bacteria and their relationships with complicated environmental factors should also be considered to further understand the functioning of these denitrifying genes pertaining to soil.

Seasonal changes in the soil microbial community as well as nitrogen-cycle gene (*nir*K) abundance have been observed in agricultural, forest, and grassland ecosystems [64–66]. Statistical analysis of the DGGE banding patterns revealed significant differences of soil microbial community between samples taken in different seasons [65]. We observed unique soil bacterial community and functional genes under different groundwater depths in both the spring (March) and the fall (August). However, unique

community of functional gene *amo*A AOA was only detected under different groundwater depths in the spring. Significantly higher nitrate contents in the spring than that in the fall (Figure 2B) suggest that seasonal dynamics of nitrate contents may greatly affect both nitrifying and denitrifying communities. In a manipulating precipitation study in beech and conifer forest plots, decreased soil water content and their effects on the total bacterial community structure were negligible, but significant effects for the active bacteria were observed during growing season [32]. The observed different microbial community structure and nitrogen functional genes of the present study may be related to differences in plant growth, belowground allocation of carbon, and obviously higher nitrate contents in the spring compared to the fall. The functional genes related to nitrification/denitrification are critical in regulating soil N cycling but account for only a small portion of the total bacterial population, such as 0.5% to 5% of the total bacterial for denitrifying microorganisms [8,67]. A relationship between seasonal temperature changes and the number of psychrophylic and mesophylic isolates was observed from the sediment–water interface [68]. Significantly different soil microbial communities and functional genes involved in nitrogen cycling were observed in spring and fall in our study, however, other parameters, such as water content, humidity and nutrient supply in soils, and their seasonal fluctuations on microbial communities and especially the nitrification/denitrification functional genes under different groundwater depths could also play a crucial role for the biogeochemical cycling processes and woody biomass production in plantation forests.

## 5. Conclusions

Soil fungal communities remained similar between tree species, whereas, soil bacterial communities are determined by groundwater depth regardless of season. Groundwater depth only affected soil archaeal communities in the spring. Soil microbial functional genes involved in nitrogen cycling varied with respect to groundwater depth. The nitrifying bacteria determined by archaeal *amo*A (AOA) nitrifying gene, and denitrifying bacteria through *nir*K genes, varied significantly with respect to groundwater depths. The tree species did not have an effect on the bacterial/archaeal and nitrifying/denitrifying microbial communities of neither the spring nor the fall samples, thus groundwater depth overrides tree-species effects on the structure of soil microbial communities involved in nitrogen cycling in plantation forests. Understanding how different nitrifying/denitrifying genes affect the function of nitrogen cycling and which stages of the nitrification/denitrification process they are involved in, through targeting respective enzymes nitrite reductase (*nir*S and *nir*K) and nitrous oxide reductase (*nos*Z), could be a key step to improve plant production and sustain our environment, such as by reducing $N_2O$ emissions.

**Author Contributions:** T.W. and D.A. conceived and designed the experiments; T.W., A.G., G.L., H.K., B.E. and D.A. performed the experiments; T.W., A.G., G.L., H.K., B.E. and D.A. analyzed the data; T.W., A.G., G.L., H.K., B.E. and D.A. contributed reagents/materials/analysis tools; T.W., A.G., G.L., H.K., B.E. and D.A. wrote the paper. All authors have read and agree to the published version of the manuscript.

**Funding:** This research was supported by the Georgia Southern University Faculty Research Grant awarded to Tiehang Wu and by the USDA National Institute of Food and Agriculture, Agriculture and Food Research Initiative [grant numbers 2013-67009-21405, 2013-67009-25148, and 2019-67019-29906] and was based upon work supported by the Department of Energy to the University of Georgia Research Foundation [grant number DE-EM0004391] and to the U.S. Forest Service Savannah River [grant number DE-EM0003622].

**Acknowledgments:** Thanks to Scott Harrison for assistance in using ABI 3500. We also thank Georgia Research for Academic Partnership in Engineering (GRAPE) Project supporting on the microbial nitrification and denitrification studies and the Department of Biology Charles Chandler Foundation of Georgia Southern University for financial support. We thank M. Bitew for preparing Figure 1 and Garret Strickland for help on optimizing DGGE conditions for nitrogen functional gene analyses.

**Conflicts of Interest:** The authors declare that research was conducted in the absence of any commercial or financial relationship that could be constructed as a potential conflict of interest.

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
