# Peer review of "Groundwater Depth Overrides Tree-Species Effects on the Structure of Soil Microbial Communities Involved in Nitrogen Cycling in Plantation Forests"

_forests, doi:10.3390/f11030275_

Round 1
Reviewer 1 Report
The manuscript is overall well written, but I feel result discussion on environmental factors in PCO ordination are confusing. For example, the red circle in the PCO ordination (Figure 2- 4) was not described. And why different environmental factors were included in Figure 2 to 4. Description for that should be included.
The author concluded a positive influence of soil organic matter to the separation of soil fungal communities in both the spring and fall- is this impact significant (Figure 2 b, c)? similarly, this question applies to Figure 3 and 4. In line 251-253, it is hard to understand the conclusion from the figure. These major results need better description and explanation.
Author Response
The manuscript is overall well written, but I feel result discussion on environmental factors in PCO ordination are confusing. For example, the red circle in the PCO ordination (Figure 2- 4) was not described. And why different environmental factors were included in Figure 2 to 4. Description for that should be included.
Thanks for the comment. The PCO ordination was further clarified in the Material and Methods part, line 196-197; the Results part, line 231-244; as well as the Figure Captions of Figure 2, 3, and 4. When all of the ten environmental factors presenting in the PCO figure, overlapping of the environment vectors will make the figure hard to read. So we only presented the absolute correlation ρ value among the top four rankings (the circle represents a correlation value of ρ = 1.0) on the figures. These have been explained in the revised manuscript.
The author concluded a positive influence of soil organic matter to the separation of soil fungal communities in both the spring and fall- is this impact significant (Figure 2 b, c)? similarly, this question applies to Figure 3 and 4. In line 251-253, it is hard to understand the conclusion from the figure. These major results need better description and explanation.
We have revised the PCO results. Indeed, soil organic matter (%) had a strong negative relationship with PCO1 to the separation, ranking among the top four based on the absolute correlation ρ value. These have been revised in the results (line 232 -235) and figure captions.
Reviewer 2 Report
The authors are representing an interesting manuscript. They are adopting a novel approach to soil microbiology where archea are playing a significant role. The papers by Daims, Schleper and other are duly acknowledged.
Yet, the data could even be better presented. I have several suggestions:
Avoid abbreviations in titles and section headings (e.g. line 145). The statement on lines 91-93 (differences in non-native / native trees) is referenced but an explanation would be helpful. Table 2: the information on sampling depth should be part of the table. The information on 0-20 cm is only given in the text of chapter 2.1.-- It is difficult to understand why CEC should change with season. the data gives doubts whether the sampled soils are indeed comparable. -- the table contains statistical interactions on interaction terms. A short narrative would be helpful. Figures 2 and 3 are insufficiently described. A clear statement is required what the reader is supposed to see. The figures themselves are not self-explanatory. The Discussion starts with 'Authors' (line 340). I suspect there was sloppy editing involved. the caption for table 4 is not sufficiently clear. the authors could aim at providing sufficient information in the caption to understand the table as a stand-alone. the measurement of soil water content is not clear. The parameter is of high relevance for the interpretation. Yet, in section 2.1 there is no mentioning of the sampling frequency. Line 365 even makes the statement that no measurements have been done. The authors should be quite clear, whether they have measured water content OR why they are convinced that the measurement was not required. the ref 20 starts with 'ubry'. Is this the right name? the Discussion is not convincing. The reader does not receive sufficient information about the extent of the observed differences (i.e. is a difference in a certain marker somewhat / highly relevant and what should be done with the information. Are there any relevant changes in biogeochemistry. -- A thorough re-work of the Discussion is necessary.
Author Response
The authors are representing an interesting manuscript. They are adopting a novel approach to soil microbiology where archea are playing a significant role. The papers by Daims, Schleper and other are duly acknowledged.
Thanks for the positive comments on the manuscript.
Yet, the data could even be better presented. I have several suggestions:
Avoid abbreviations in titles and section headings (e.g. line 145).
This has been corrected.
The statement on lines 91-93 (differences in non-native / native trees) is referenced but an explanation would be helpful.
It has been further explained in line 64-65.
Table 2: the information on sampling depth should be part of the table. The information on 0-20 cm is only given in the text of chapter 2.1.
This has been added in the Table 2.
-- It is difficult to understand why CEC should change with season. the data gives doubts whether the sampled soils are indeed comparable.
This is a great point. We didn’t explain this very well in the manuscript. After checking the data closely, it can be seen that the soil nitrate, phosphorus, magnesium and potassium concentration are significantly different with season as well. The root growth and activities should be different, therefor affect the nutrient uptakes differently and let to different nutrient concentration and CEC with season. This has been added in the revised manuscript Line 365 - 368.
-- the table contains statistical interactions on interaction terms. A short narrative would be helpful.
The interaction between Tree and Season for the phosphorus and magnesium was detected, suggesting the different root growth and activities with season as well as different characteristics of pine and eucalyptus roots could affect the nutrient uptake differently, therefore let to the observed interaction on soil phosphorus and magnesium concentration. This has been added in the revised manuscript Line 368 - 371.
Figures 2 and 3 are insufficiently described. A clear statement is required what the reader is supposed to see. The figures themselves are not self-explanatory.
Thanks for the comments. It is pointed out by the reviewer #1. The PCO ordination was further clarified in the Material and Methods part, line 196-197; the Results part, line 231-244; as well as the Figure Captions of Figure 2, 3, and 4.
The Discussion starts with 'Authors' (line 340). I suspect there was sloppy editing involved.
It has been corrected.
the caption for table 4 is not sufficiently clear. the authors could aim at providing sufficient information in the caption to understand the table as a stand-alone.
It has been improved based on the suggestion.
the measurement of soil water content is not clear. The parameter is of high relevance for the interpretation. Yet, in section 2.1 there is no mentioning of the sampling frequency. Line 365 even makes the statement that no measurements have been done. The authors should be quite clear, whether they have measured water content OR why they are convinced that the measurement was not required.
It has been revised. It should be “The spring and fall water content was not significantly different in the collected soil samples of two sampling time at present study, however, the seasonal fluctuation of the water content and the effects of anoxic conditions of the soils for the effect of anaerobic soils may cause the seasonal difference between the functional genes.” Line 379-382.
the ref 20 starts with 'ubry'. Is this the right name?
It has been corrected.
the Discussion is not convincing. The reader does not receive sufficient information about the extent of the observed differences (i.e. is a difference in a certain marker somewhat / highly relevant and what should be done with the information. Are there any relevant changes in biogeochemistry. -- A thorough re-work of the Discussion is necessary.
We have made revision on the discussion and emphasized the understanding factors on microbial communities and especially the nitrification/denitrification functional genes could also play a crucial role for the biogeochemical cycling processes and woody biomass production in plantation forests (line 469 – 474).
Reviewer 3 Report
Dear Editor,
I carefully read the manuscript entitled “Groundwater depth overrides tree-species effects on the structure of soil microbial communities involved in nitrogen cycling in plantation forests” from Wu et al. The objectives of the study are clear and experimental design is appropriate although the methods used to investigate the microbial communities of soil are not update. In fact, the molecular approaches used (i.e. DGGE) have fundamental technical limitations, as demonstrated by the fact that several functional genes (bacterial amoA, nrxA, narG and nirS) are not detected. It is hardly to support that all these genes are not present in the soil.
However, my major concern regard the topic and I don’t think that the paper may be well placed in the special issue “Ecological Insights into the Sustainable Development of Bioenergy from Forests” since there is no data and discussion concerning the assessment of the “ecological benefits, costs, and consequences associated with using woody biomass for bioenergy”. It is well clear in the special issue information that the request of understanding the influence of different vegetation, air, soil, and water regimes is related to woody biomass production and not to soil microbial communities. For this reason, I recommend rejection, suggesting to the authors to improve their study and to redirect the paper to another special issue.
Author Response
I carefully read the manuscript entitled “Groundwater depth overrides tree-species effects on the structure of soil microbial communities involved in nitrogen cycling in plantation forests” from Wu et al. The objectives of the study are clear and experimental design is appropriate although the methods used to investigate the microbial communities of soil are not update. In fact, the molecular approaches used (i.e. DGGE) have fundamental technical limitations, as demonstrated by the fact that several functional genes (bacterial amoA, nrxA, narG and nirS) are not detected. It is hardly to support that all these genes are not present in the soil.
We agree on this point. Each molecular method has its weakness for understanding the real soil microbial communities. We didn’t detect several functional genes by regular DGGE techniques, such as nitrite oxidoreductase genes nrxA of nitrite oxidizer. This gene was detected by a two-step nested PCR reaction, suggesting the low amounts of the nrxA gene in soils. Using Q-PCR, this was further supported. We think this needs to be further investigated in the future by next generation sequencing or other molecular techniques, and it is an ongoing project we are working on.
However, my major concern regard the topic and I don’t think that the paper may be well placed in the special issue “Ecological Insights into the Sustainable Development of Bioenergy from Forests” since there is no data and discussion concerning the assessment of the “ecological benefits, costs, and consequences associated with using woody biomass for bioenergy”. It is well clear in the special issue information that the request of understanding the influence of different vegetation, air, soil, and water regimes is related to woody biomass production and not to soil microbial communities. For this reason, I recommend rejection, suggesting to the authors to improve their study and to redirect the paper to another special issue.
Thanks for the comments and suggestions. We have modified the discussion and emphasized on the soil microbial community and nitrogen cycling functional genes will potentially play a crucial role for the biogeochemical cycling processes and woody biomass production in plantation forests (line 469 – 474).
We also appreciate that the journal Managing Editor think the article is probably fine for this special issue, but with need to address the concerns by the reviewers. We have made revisions on the manuscript based on the comments and suggestions of reviewers we could. We hope it could be accepted for publication on the special issue “Ecological Insights into the Sustainable Development of Bioenergy from Forests”.
Round 2
Reviewer 2 Report
The authors have addressed the assessment of the reviewers of an earlier version and have improved the paper accordingly. I recommend its publication.
Reviewer 3 Report
I continue to remain of my opinion that the manuscript is not suitable for the special issue "Ecological insights into the sustainable development of bioenergy from forests" since it does not deal with the topic on which the issue is focused. I think that adding a sentence regarding the microorganisms that “will potentially play a crucial role for the biogeochemical cycling processes and woody biomass production in plantation forests (lines 469 – 474 it is not in keeping with special issue information. Therefore, despite the improvement of the manuscript due to a good work of revision, I continue in my idea to reject the manuscript and to ask to the journal to redirect it to another special issue.